# Limited-Memory Greedy Quasi-Newton Method with Non-asymptotic Superlinear Convergence Rate

## Abstract

Non-asymptotic convergence analysis of quasi-Newton methods has gained attention with a landmark result establishing an explicit local superlinear rate of $\mathcal{O}((1/\sqrt{t})^t)$. The methods that obtain this rate, however, exhibit a well-known drawback: they require the storage of the previous Hessian approximation matrix or instead storing all past curvature information to form the current Hessian inverse approximation. Limited-memory variants of quasi-Newton methods such as the celebrated L-BFGS alleviate this issue by leveraging a limited window of past curvature information to construct the Hessian inverse approximation. As a result, their per iteration complexity and storage requirement is $\mathcal{O}(\tau d)$ where $\tau \leq d$ is the size of the window and $d$ is the problem dimension reducing the $\mathcal{O}(d^2)$ computational cost and memory requirement of standard quasi-Newton methods. However, to the best of our knowledge, there is no result showing a non-asymptotic superlinear convergence rate for any limited-memory quasi-Newton method. In this work, we close this gap by presenting a Limited-memory Greedy BFGS (LG-BFGS) method that can achieve an explicit non-asymptotic superlinear rate. We incorporate displacement aggregation, i.e., decorrelating projection, in post-processing gradient variations, together with a basis vector selection scheme on variable variations, which *greedily* maximizes a progress measure of the Hessian estimate to the true Hessian. Their combination allows past curvature information to remain in a sparse subspace while yielding a valid representation of the full history. Interestingly, our established *non-asymptotic* superlinear convergence rate demonstrates an explicit trade-off between the convergence speed and memory requirement, which to our knowledge, is the first of its kind. Numerical results corroborate our theoretical findings and demonstrate the effectiveness of our method.

## 1 Introduction

This work focuses on the minimization of a smooth and strongly convex function as follows:

$$\min_{\mathbf{x}} f(\mathbf{x}) \tag{1}$$

where $\mathbf{x} \in \mathbb{R}^d$ and $f : \mathbb{R}^d \to \mathbb{R}$ is twice differentiable. It is well-known that gradient descent and its accelerated variant can converge to the optimal solution at a linear rate (Nesterov, 2003). While this is advantageous from the perspective of simplicity and low complexity of $\mathcal{O}(d)$, the convergence rate for these methods exhibits dependence on the conditioning of the problem. Therefore, when the problem is ill-conditioned, such methods could be slow. Second-order methods such as Newton's method or cubic regularized Newton method are able to address the ill-conditioning issue by leveraging the objective function curvature. Alas, their complexity of $\mathcal{O}(d^3)$ limits their application.

Quasi-Newton techniques strike a balance between gradient methods and second-order algorithms in terms of having a complexity of $\mathcal{O}(d^2)$, while refining the dependence on the problem conditioning and achieving local superlinear rates (Byrd et al., 1987; Gao & Goldfarb, 2019). They approximate the Hessian or the Newton direction in a variety of ways leading to different algorithms, such as Broyden's method (Broyden, 1965; Broyden et al., 1973; Gay, 1979), the celebrated Broyden (Broyden, 1967; Broyden et al., 1973)-Fletcher(Fletcher & Powell, 1963)-Goldfarb(Goldfarb,

1970)-Shanno(Shanno, 1970) (BFGS) method, as well as Davidon-Fletcher-Powell (DFP) algorithm (Fletcher & Powell, 1963; Davidon, 1991) and Symmetric Rank 1 (SR1) method (Toint, 1991). For some time, it has been known that these methods can achieve asymptotic local superlinear rates (Dennis & Moré, 1974). More recently, the explicit non-asymptotic local rate has been characterized for Broyden class when combined with a greedy basis vector selection (Rodomanov & Nesterov, 2021a; Lin et al., 2021), and subsequently, the explicit rates have been derived for standard BFGS (Rodomanov & Nesterov, 2021b; Jin & Mokhtari, 2022) and SR1 (Ye et al., 2023). Additionally, Nesterov acceleration has been shown to enhance the convergence rate of standard BFGS after a threshold number of iterations under suitable choice of step-size and error-bound tolerance (Sahu & Pattanaik, 2023a), and (Jiang et al., 2023) provides an analysis methodology for both local and global superlinear convergence. While these results are promising, there exist important limitations. In particular, their per-update complexity is quadratic $\mathcal{O}(d^2)$ which could be costly in high-dimensional settings. Also, they require to store either the most recent Hessian inverse approximation on a memory of $\mathcal{O}(d^2)$ or all past differences of gradients and iterates, known as curvature information, on a memory of $\mathcal{O}(td)$. Hence, in the case that the iteration index $t$ becomes large, their storage requirement becomes quadratic in dimension.

Limited-memory BFGS (L-BFGS) drops all but the past $\tau$ curvature pairs, alleviating these bottlenecks and reducing the iteration complexity and memory requirement to $\mathcal{O}(\tau d)$. Note that $\tau$ is often a constant much smaller than the dimension $d$. However, previous analyses of classic L-BFGS only guarantee either a sublinear (Mokhtari & Ribeiro, 2015; Boggs & Byrd, 2019; Yousefian et al., 2020) or a linear (Moritz et al., 2016) convergence rate, which are no better than gradient descent. Recently, Berahas et al. (2022a) showed that when L-BFGS is combined with a way to project removed information onto the span that remains, it asymptotically converges at a superlinear rate. However, they failed to characterize the region for which the superlinear rate is achieved, the explicit rate of convergence, or the role of memory size. Thus, we pose the following three questions:

> *Is it possible for fixed-size limited-memory methods to achieve explicit superlinear rates? If so, what condition does it require? What is the explicit effect of memory size on the convergence rate?*

**Contributions.** To answer these questions, we summarize our contributions as follows:

**1.** We propose a Limited-memory Greedy (LG)-BFGS method, which is a novel synthesis of greedy basis selection and displacement aggregation. It maximizes the update progress per iteration to improve the convergence, while maintaining the limited memory setting.

**2.** We furnish an explicit expression of the non-asymptotic convergence for LG-BFGS, which delineates the direct influence of memory size through the concept of relative condition number $\beta_\tau$ and identifies roles played by other factors (e.g., problem settings, initial conditions). This is the first work that characterizes the memory effect on the convergence rate.

**3.** LG-BFGS presents a superlinear rate when assuming $\beta_\tau$ is bounded. This is the first work that shows the possibility of achieving superlinear rates in limited-memory methods and identifies the exact condition under which this is possible. The result recovers that of the full-memory method when memory size increases to feature dimension. This establishes a clear path from limited-memory methods to full-memory ones, and explains the performance change along the memory reduction.

In this work, we focus on the BFGS method in the class of quasi-Newton methods, while similar techniques could be applied to its alternative variants.

**Related Work.** Alternative approach for memory reduction is randomized sub-sampling (sketching) of Hessian matrices (Pilanci & Wainwright, 2017; Gower & Richtárik, 2017; Gower et al., 2019). This distinct line of inquiry can establish local quadratic rates of convergence when the memory associated with the sub-sampling size achieves a threshold called effective problem dimension (Lacotte et al., 2021); however, ensuring this condition in practice is elusive, which leads to nontrivial parameter tuning issues in practice (Derezinski et al., 2021). Efficient approximation criteria for sketching is an ongoing avenue of research, see e.g., (Derezinski et al., 2020).

**Notation.** The weighted norm of $\mathbf{x}$ by a positive definite matrix $\mathbf{Z}$ is denoted by $\|\mathbf{x}\|_{\mathbf{Z}} = \sqrt{\mathbf{x}^\top \mathbf{Z} \mathbf{x}}$.

## 2 PROBLEM SETUP AND BACKGROUND

This paper focuses on quasi-Newton (QN) algorithms for problem 1, which aim at approximating the objective function curvature. Let $\mathbf{x}_t$ be the decision variable, $\nabla f(\mathbf{x}_t)$ the gradient, and $\mathbf{B}_t$ the

Hessian approximation at iteration $t$. The update rule for most QN methods follows the recursion

$$\mathbf{x}_{t+1} = \mathbf{x}_t - \alpha \mathbf{H}_t \nabla f(\mathbf{x}_t) \tag{2}$$

where $\alpha$ is the step-size and $\mathbf{H}_t$ is the inverse of the Hessian approximation matrix $\mathbf{B}_t$. By updating $\mathbf{B}_t$ with different schema, one may obtain a variety of QN methods (Davidon, 1991). In the case of Broyden-Fletcher-Goldfarb-Shanno (BFGS), we define the variable variation $\mathbf{s}_t = \mathbf{x}_{t+1} - \mathbf{x}_t$ and gradient variation $\mathbf{r}_t = \nabla f(\mathbf{x}_{t+1}) - \nabla f(\mathbf{x}_t)$, and sequentially select the matrix close to the previous estimate, while satisfying the secant condition $\mathbf{B}\mathbf{s}_t = \mathbf{r}_t$. This would lead to the following update

$$\mathbf{B}_{t+1} = \mathbf{B}_t + \frac{\mathbf{r}_t \mathbf{r}_t^\top}{\mathbf{r}_t^\top \mathbf{s}_t} - \frac{\mathbf{B}_t \mathbf{s}_t \mathbf{s}_t^\top \mathbf{B}_t^\top}{\mathbf{s}_t^\top \mathbf{B}_t \mathbf{s}_t}. \tag{3}$$

Given the rank-two structure of the above update, one can use the Sherman-Morrison formula to establish a simple update for the Hessian inverse approximation matrix which is given by

$$\mathbf{H}_{t+1} = \left(\mathbf{I} - \frac{\mathbf{r}_t \mathbf{s}_t^\top}{\mathbf{s}_t^\top \mathbf{r}_t}\right)^\top \mathbf{H}_t \left(\mathbf{I} - \frac{\mathbf{r}_t \mathbf{s}_t^\top}{\mathbf{s}_t^\top \mathbf{r}_t}\right) + \frac{\mathbf{s}_t \mathbf{s}_t^\top}{\mathbf{s}_t^\top \mathbf{r}_t}. \tag{4}$$

The computational cost of this update is $\mathcal{O}(d^2)$ as it requires matrix-vector product calculations. Moreover, it requires a memory of $\mathcal{O}(d^2)$ to store the matrix $\mathbf{H}_t$. Using the recursive structure of the update in equation 4 one can also implement BFGS at the computational cost and memory of $\mathcal{O}(td)$ which is beneficial in the regime that $t \leq d$, while this gain disappears as soon as $t > d$. More precisely, suppose we are at iteration $t$ and we have access to all previous iteration and gradient differences $\{(\mathbf{s}_u, \mathbf{r}_u)\}_{u=0}^{t-1}$, referred to as the *curvature pairs*, as well as the initial Hessian inverse approximation matrix $\mathbf{H}_0$ that is diagonal. The memory requirement for storing the above information is $\mathcal{O}(td)$. It can be shown that the BFGS descent direction $-\mathbf{H}_t \nabla f(\mathbf{x}_t)$ can be computed at an overall cost of $\mathcal{O}(td)$ with the well-known two-loop recursion (Nocedal, 1980). Hence, one can conclude that the computational cost and storage requirement of BFGS is $\mathcal{O}(\min(td, d^2))$.

The above observation motivates the use of limited-memory QN methods and in particular limited memory BFGS (L-BFGS), in which we only keep last $\tau$ pairs of curvature information, i.e., $\{(\mathbf{s}_u, \mathbf{r}_u)\}_{u=t-\tau}^{t-1}$, to construct the descent direction at each iteration. If $\tau < t$ and $\tau < d$ it will reduce the computational cost and storage requirement of BFGS from $\mathcal{O}(\min(td, d^2))$ to $\mathcal{O}(\tau d)$. Indeed, a larger $\tau$ leads to a better approximation of the BFGS descent direction but higher computation and memory, while a smaller $\tau$ reduces the memory and computation at the price of having a less accurate approximation of the BFGS direction. A key observation is that for such limited memory updates the standard analysis of the Hessian approximation error does not go through, and hence the theoretical results for standard QN methods (without limited memory) may not hold anymore. To the best of our knowledge, all existing analyses of L-BFGS can only provide linear convergence rates, which is not better than the one for gradient descent (Liu & Nocedal, 1989; Mokhtari & Ribeiro, 2015; Moritz et al., 2016; Bollapragada et al., 2018; Berahas & Takáč, 2020; Berahas et al., 2022b). Therefore, the question is how to design a memory-retention procedure so that a non-asymptotic superlinear rate can be achieved in a limited-memory manner. Doing so is the focus of the following section.

## 3   Limited-Memory Greedy BFGS (LG-BFGS)

We propose a new variant of L-BFGS that departs from the classical approach in two key ways: (i) greedy basis vector selection for the variable variation and (ii) displacement aggregation on the gradient variation. Greedy updates have been introduced into QN methods as a way to maximize a measure of progress, the trace Lyapunov function, w.r.t. variable variation, which enables to derive an explicit per-step improvement of the Hessian approximation to the true Hessian. The upshot of this modification is an explicit (non-asymptotic) superlinear convergence rate (Rodomanov & Nesterov, 2021a). Moreover, as we will see, the basis vector selection of the greedy update allows to control the memory growth, which is not present in classical QN schemes and provides a quantitative motivation to port greedy updates (Rodomanov & Nesterov, 2021a) into the limited-memory setting.

However, greedy selection alone with respect to variable variation cannot ensure that the resultant L-BFGS retains the correct curvature information, as L-BFGS stores only a limited number of past gradient values. Displacement aggregation (Berahas et al., 2022a) provides a way to cull linearly

dependent curvature information such that a retained subset gradient variation information is sufficient for ensuring the quality of the Hessian inverse approximation, which can result in asymptotic superlinear rates (Berahas et al., 2022a; Sahu & Pattanaik, 2023b). We synthesize these two key strategies in a unique fashion and present our Limited-memory Greedy BFGS (LG-BFGS) method.

Let $\mathbf{x}_0$ be the initial variable, $\nabla f(\mathbf{x}_0)$ the initial gradient, and $\mathbf{B}_0$ the initial Hessian approximation satisfying $\mathbf{B}_0 \succeq \nabla^2 f(\mathbf{x}_0)$, which can be easily satisfied by setting $\mathbf{B}_0 = L\mathbf{I}$. The LG-BFGS consists of two phases at each iteration: (i) decision variable update and (ii) curvature pair update.

**Decision variable update:** At iteration $t$, we have access to the initial Hessian approximation, previously stored curvature pairs, the number of which is at most $\tau$, the current iterate $\mathbf{x}_t$ and its gradient $\nabla f(\mathbf{x}_t)$. Similar to L-BFGS, we can compute the descent direction of LG-BFGS denoted by $\mathbf{d}_t$ at a cost of at most $\mathcal{O}(\tau d)$ and then find the new iterate with stepsize $\alpha$ as

$$\mathbf{x}_{t+1} = \mathbf{x}_t + \alpha \mathbf{d}_t. \tag{5}$$

**Curvature pair update:** The update of the curvature pairs happens in two steps as described below.

*(1) Greedy step* selects the curvature pair greedily to maximize the update progress of the Hessian approximation. Specifically, it selects the variable variation $\mathbf{s}_t$ from a vector basis $\{\mathbf{e}_1, \ldots, \mathbf{e}_d\}$ by maximizing the ratio $\frac{\mathbf{s}^\top \mathbf{B}_t \mathbf{s}}{\mathbf{s}^\top \nabla^2 f(\mathbf{x}_{t+1})\mathbf{s}}$, where $\{\mathbf{e}_1, \ldots, \mathbf{e}_d\}$ can be any basis in the space. For ease of implementation, a reasonable choice is selecting $\mathbf{e}_i$ as the vector that its $i$-th entry is the only non-zero value. For the purpose of limited memory, we propose an innovative search scheme that restricts the vector basis to the subset $\{\mathbf{e}_1, \ldots, \mathbf{e}_\tau\}$ of $\tau < d$ vectors, and select $\mathbf{s}_t$ greedily as

$$\mathbf{s}_t = \underset{\mathbf{s} \in \{\mathbf{e}_1, \ldots, \mathbf{e}_\tau\}}{\arg \max} \frac{\mathbf{s}^\top \mathbf{B}_t \mathbf{s}}{\mathbf{s}^\top \nabla^2 f(\mathbf{x}_{t+1})\mathbf{s}}. \tag{6}$$

The greedy step selects $\mathbf{s}_t$ such that it maximizes the progress of Hessian approximation. For the creation of our curvature pair, instead of using the classic $\mathbf{x}_{t+1} - \mathbf{x}_t$ as the variable variation, we use the output $\mathbf{s}_t$ of equation 6. Note that equation 6 is equivalent to computing $\tau$ diagonal components of the matrix $\mathbf{B}_t$ and $\nabla^2 f(\mathbf{x}_{t+1})$, which can be done efficiently at a cost of $\mathcal{O}(\tau d)$, and matrix-vector products $\{\mathbf{B}_t \mathbf{e}_i\}_{i=1}^\tau$ can be computed with a compact representation at a cost of $\mathcal{O}(\tau^2 d)$ without forming $\mathbf{B}_t$; see Section 7.2 in (Nocedal & Wright, 2006). This is due to the fact that $\mathbf{B}_t$ only depends on $\tau$ curvature pairs. Once $\mathbf{s}_t$ is computed, we can efficiently compute $\mathbf{r}_t$ by following $\mathbf{r}_t = \nabla^2 f(\mathbf{x}_{t+1})\mathbf{s}_t$. Since $\mathbf{s}_t$ is a unit vector with only one non-zero entry, we can simply compute $\mathbf{r}_t$ by adding the elements of one column of $\nabla^2 f(\mathbf{x}_{t+1})$. Hence, all computations are $\mathcal{O}(d)$.

*(2) Displacement step.* Given access to $\hat{\tau}$ previously stored curvature pairs $\mathcal{P}_{t-1} = \{(\mathbf{s}_u, \mathbf{r}_u)\}_{u=0}^{\hat{\tau}-1}$, where $\hat{\tau}$ is either less than or equal to our memory budget, i.e., $\hat{\tau} \leq \tau$, and a new curvature pair $(\mathbf{s}_t, \mathbf{r}_t)$ from the evaluation of equation 6, the key question is how to select which information to retain for curvature pair update. Displacement aggregation incorporates $(\mathbf{s}_t, \mathbf{r}_t)$ into $\mathcal{P}_{t-1}$ and formulates an aggregated curvature set $\mathcal{P}_t$ rather than simply removing the oldest pair $(\mathbf{s}_0, \mathbf{r}_0)$ and adding the latest one $(\mathbf{s}_t, \mathbf{r}_t)$. Doing so yields a limited-memory Hessian approximation equal to that computed from a full-memory version; hence, reducing the effect of memory reduction and improving the convergence rate. The execution of this retention strategy contains three cases:

**(C1) If new variable variation $\mathbf{s}_t$ is linearly independent of $\{\mathbf{s}_u\}_{u=0}^{\hat{\tau}-1}$ in $\mathcal{P}_{t-1}$**, we formulate $\mathcal{P}_t$ by directly adding the new curvature pair as

$$\mathcal{P}_t = \{\mathcal{P}_{t-1}, (\mathbf{s}_t, \mathbf{r}_t)\}. \tag{7}$$

**(C2) If new variable variation $\mathbf{s}_t$ is parallel to last variable variation $\mathbf{s}_{\hat{\tau}-1}$ in $\mathcal{P}_{t-1}$**, we formulate $\mathcal{P}_t$ by removing the most recently stored $(\mathbf{s}_{\hat{\tau}-1}, \mathbf{r}_{\hat{\tau}-1})$ and replace it with the new $(\mathbf{s}_t, \mathbf{r}_t)$ as

$$\mathcal{P}_t = \{(\mathbf{s}_0, \mathbf{r}_0), \ldots, (\mathbf{s}_{\hat{\tau}-2}, \mathbf{r}_{\hat{\tau}-2}), (\mathbf{s}_t, \mathbf{r}_t)\}. \tag{8}$$

I.e., we have $(\mathbf{s}_{\hat{\tau}-1}, \mathbf{r}_{\hat{\tau}-1}) = (\mathbf{s}_t, \mathbf{r}_t)$ in the updated $\mathcal{P}_t$.

**(C3) If new variable variation $\mathbf{s}_t$ is parallel to a previously stored variable variation in $\mathcal{P}_{t-1}$**, we formulate $\mathcal{P}_t$ by projecting the information of $(\mathbf{s}_t, \mathbf{r}_t)$ onto the previous pairs to modify and reweigh $\mathcal{P}_{t-1}$. Specifically, assume $\mathbf{s}_t = \mathbf{s}_j$ for some $0 \leq j < \hat{\tau} - 1$. Denote by $\mathbf{S}_{j_1:j_2} = [\mathbf{s}_{j_1} \cdots \mathbf{s}_{j_2}] \in \mathbb{R}^{d \times (j_2 - j_1 + 1)}$ and $\mathbf{R}_{j_1:j_2} = [\mathbf{r}_{j_1} \cdots \mathbf{r}_{j_2}] \in \mathbb{R}^{d \times (j_2 - j_1 + 1)}$ the concatenated variable variation matrix

---

**Algorithm 1** Limited-memory Greedy BFGS (LG-BFGS) Method

---

**Initialization:** Loss function $f(\mathbf{x})$, initial decision vector $\mathbf{x}_0$ and Hessian inverse approximation $\mathbf{H}_0$
**for** $t = 0, 1, \ldots, T$ **do**
1. Compute the gradient $\nabla f(\mathbf{x}_t)$ and update the decision variable $\mathbf{x}_{t+1}$ as in equation 5
2. Compute $\phi_t = \|\mathbf{x}_{t+1} - \mathbf{x}_t\|_{\nabla^2 f(\mathbf{x}_t)}$, the scale constant $\psi_t = 1 + C_M \phi_t$, the corrected initial $\mathbf{H}_0 = \psi_t^{-1} \mathbf{H}_0$ and the scaled curvature pairs $\{\mathbf{s}_u, \tilde{\mathbf{r}}_u\}_{u=0}^{\hat{\tau}-1}$ with $\tilde{\mathbf{r}}_u = \psi_t \mathbf{r}_u$
3. Select the greedy curvature pair $(\mathbf{s}_t, \mathbf{r}_t)$ as in equation 6 with the corrected curvature pairs $\{\mathbf{s}_u, \tilde{\mathbf{r}}_u\}_{u=0}^{\hat{\tau}-1}$
4. Update the historical curvature pairs $\mathcal{P}_t$ as in equation 7-equation 9

---

and gradient variation matrix for $0 \leq j_1 \leq j_2$, and define $(\mathbf{s}_{\hat{\tau}}, \mathbf{r}_{\hat{\tau}}) = (\mathbf{s}_t, \mathbf{r}_t)$ as the new curvature pair. We remove $(\mathbf{s}_j, \mathbf{r}_j)$, replace $\mathbf{R}_{j+1:\hat{\tau}}$ with a modified $\hat{\mathbf{R}}_{j+1:\hat{\tau}}$, and formulate $\mathcal{P}_t$ as

$$\mathcal{P}_t = \{(\mathbf{s}_0, \mathbf{r}_0), \ldots, (\mathbf{s}_{j-1}, \mathbf{r}_{j-1}), (\mathbf{s}_{j+1}, \hat{\mathbf{r}}_{j+1}), \ldots, (\mathbf{s}_{\hat{\tau}}, \hat{\mathbf{r}}_{\hat{\tau}})\} \tag{9}$$

such that the descent direction in equation 5 computed from $\mathcal{P}_t$ is equal to that from $\{\mathcal{P}_{t-1}, (\mathbf{s}_{\hat{\tau}}, \mathbf{r}_{\hat{\tau}})\}$. To do so, we compute $\hat{\mathbf{R}}_{j+1:\hat{\tau}} = [\hat{\mathbf{r}}_{j+1} \cdots \hat{\mathbf{r}}_{\hat{\tau}}]$ following the aggregation procedure in (Berahas et al., 2022a) as

$$\hat{\mathbf{R}}_{j+1:\hat{\tau}} = (\mathbf{H}_{0:j-1})^{-1} \mathbf{S}_{j+1:\hat{\tau}} [\mathbf{A} \; \mathbf{0}] + \mathbf{r}_j [\mathbf{b} \; \mathbf{0}] + \mathbf{R}_{j+1:\hat{\tau}} \tag{10}$$

where $\mathbf{H}_{0:j-1}$ is the Hessian inverse approximation computed with the limited-memory strategy, initialized at $\mathbf{H}_0$ and updated with $j$ curvature pairs $\{(\mathbf{s}_u, \mathbf{r}_u)\}_{u=0}^{j-1}$, and $\mathbf{b} \in \mathbb{R}^{\hat{\tau}-j-1}$ and $\mathbf{A} \in \mathbb{R}^{(\hat{\tau}-j) \times (\hat{\tau}-j-1)}$ are unknowns that can be computed by Algorithm 4 in (Berahas et al., 2022a). Note that $\mathbf{H}_{0:j-1}^{-1} \mathbf{S}_{j+1:\hat{\tau}}$ in (10) can be computed without forming $\mathbf{H}_{0:j-1}$ (Berahas et al., 2022a).

We remark that the number of curvature pairs $\hat{\tau}$ in $\mathcal{P}_t$ is always bounded by the memory size $\tau$, i.e., $\hat{\tau} \leq \tau$ for all iterations $t$. We explain this fact from two aspects:

*(i)* (C1) only happens when $\hat{\tau} < \tau$. Specifically, the variable variations $\{\mathbf{s}_u\}_{u=0}^{\hat{\tau}-1}$ in $\mathcal{P}_{t-1}$ are independent because any dependent variable variation will not be included into $\mathcal{P}_{t-1}$ according to (C2) and (C3). If $\hat{\tau} \geq \tau$, the new variable variation $\mathbf{s}_t$ cannot be linearly independent of $\{\mathbf{s}_u\}_{u=0}^{\hat{\tau}-1}$ in $\mathcal{P}_{t-1}$ because all variable variations are selected from the subset $\{\mathbf{e}_1, \ldots, \mathbf{e}_\tau\}$ of size $\tau$ [cf. equation 6]. Thus, the number of curvature pairs in $\mathcal{P}_t$ is bounded by the memory size, i.e., $\hat{\tau} + 1 \leq \tau$.

*(ii)* (C2) and (C3) do not increase the number of curvature pairs from $\mathcal{P}_{t-1}$ to $\mathcal{P}_t$ and thus, the number of curvature pairs in $\mathcal{P}_t$ is bounded by the memory size, i.e., $\hat{\tau} \leq \tau$.

Therefore, LG-BFGS stores at most $\tau$ curvature pairs and solves problem 1 with limited memory.

**Remark 1** (Computational Cost). *While the displacement step seems computationally costly, it can be implemented efficiently. For the case selection, it is computationally efficient because variable variations $\{\mathbf{s}_0, \ldots, \mathbf{s}_{\hat{\tau}-1}\}$ are selected from the same subset $\{\mathbf{e}_1, \ldots, \mathbf{e}_\tau\}$ instead of distributed arbitrarily in $d$-dimensional space and thus, the computational cost is of $\mathcal{O}(\tau)$. For the modified $\hat{\mathbf{R}}_{j+1:\hat{\tau}}$, the computational cost is of $\mathcal{O}(\tau^2 d + \tau^4)$ as analyzed in (Berahas et al., 2022a).*

## 3.1 Correction Strategy

The greedy step is designed to maximize the progress of the BFGS update in terms of the trace function $\sigma(\nabla^2 f(\mathbf{x}_{t+1}), \mathbf{B}_{t+1}) = \text{Tr}(\nabla^2 f(\mathbf{x}_{t+1})^{-1} \mathbf{B}_{t+1}) - d$, which captures the difference between positive definite matrices. For our analysis, we require this expression to stay positive and for that reason, we need the following condition $\mathbf{B}_{t+1} \succeq \nabla^2 f(\mathbf{x}_{t+1})$. Note that this condition may not hold even if $\mathbf{B}_t \succeq \nabla^2 f(\mathbf{x}_t)$, which requires developing a correction strategy to overcome this issue. Specifically, define $\phi_t := \|\mathbf{x}_{t+1} - \mathbf{x}_t\|_{L\mathbf{I}}$ where $L$ is the Lipschitz constant of the function gradient [Asm. 1] and the corrected matrix $\hat{\mathbf{B}}_t = (1 + \phi_t C_M) \mathbf{B}_t \succeq (1 + \|\mathbf{x}_{t+1} - \mathbf{x}_t\|_{\nabla^2 f(\mathbf{x}_t)} C_M) \mathbf{B}_t \succeq \nabla^2 f(\mathbf{x}_{t+1})$ as a scaled version of $\mathbf{B}_t$ where $C_M$ is the strongly self-concordant constant; see equation 13. By updating $\mathbf{B}_{t+1}$ with the corrected $\hat{\mathbf{B}}_t$, we can ensure that $\mathbf{B}_{t+1} \succeq \nabla^2 f(\mathbf{x}_{t+1})$ is satisfied (Rodomanov & Nesterov, 2021a). Note that computing $\phi_t$ only requires the computational cost $\mathcal{O}(d)$ because $L\mathbf{I}$ is a diagonal matrix. However, in the limited memory setting, we do not have access to the Hessian

approximation matrix explicitly and thus need to apply this scaling to the curvature pairs. This can be done by scaling the gradient variation as $\tilde{\mathbf{r}}_t = (1 + \phi_t C_M)\mathbf{r}_t$. In fact, this means that we need to use the corrected gradient variation $\tilde{\mathbf{r}}_t$ instead of $\mathbf{r}_t$ for the displacement step.

Next, we formally prove that using the scaled gradient variation $\tilde{\mathbf{r}}_t$ would equivalently lead to scaling the corresponding Hessian approximation matrices.

**Proposition 1.** *Consider the BFGS update in equation 3. Let $\mathbf{B}_0$ be the initial Hessian approximation, $\psi_t = 1 + \phi_t C_M$ be the scaling parameter, and $\{(\mathbf{s}_u, \mathbf{r}_u)\}_{u=0}^{t-1}$ be the curvature pairs. Define the modified curvature pairs $\{(\mathbf{s}_u, \tilde{\mathbf{r}}_u)\}_{u=0}^{t-1}$ as $\tilde{\mathbf{r}}_u = \psi_t \mathbf{r}_u$ for all $u = 0, \ldots, t-1$. At iteration t, let $\hat{\mathbf{B}}_t$ be the corrected Hessian approximation as*

$$\hat{\mathbf{B}}_t = \psi_t \mathrm{BFGS}(\mathbf{B}_{t-1}, \mathbf{s}_{t-1}, \mathbf{r}_{t-1}), \ \ldots, \ \mathbf{B}_1 = \mathrm{BFGS}(\mathbf{B}_0, \mathbf{s}_0, \mathbf{r}_0) \tag{11}$$

*where* $\mathrm{BFGS}(\cdot)$ *represents the operation in equation 3. Then, given* $\tilde{\mathbf{B}}_0 = \psi_t \mathbf{B}_0$*, it holds that*

$$\hat{\mathbf{B}}_t = \mathrm{BFGS}(\tilde{\mathbf{B}}_{t-1}, \mathbf{s}_{t-1}, \tilde{\mathbf{r}}_{t-1}), \ \ldots, \ \tilde{\mathbf{B}}_1 = \mathrm{BFGS}(\tilde{\mathbf{B}}_0, \mathbf{s}_0, \tilde{\mathbf{r}}_0). \tag{12}$$

Proposition 1 states that we can transform the correction strategy of the Hessian approximation $\mathbf{B}_t$ [cf. equation 11] to the corresponding correction strategy of the gradient variation $\mathbf{r}_t$. This indicates that we can incorporate the correction strategy into the displacement step by scaling the gradient variation $\mathbf{r}_t$ and maintain the remaining unchanged. Algorithm 1 formally summarizes the corrected LG-BFGS method which alleviates the requirement of $\mathbf{B}_{t+1} \succeq \nabla^2 f(\mathbf{x}_{t+1})$.

## 4 CONVERGENCE

In this section, we conduct convergence analysis to show that LG-BFGS can achieve an explicit superlinear rate depending on the memory size $\tau$. This result is salient as it is the first non-asymptotic superlinear rate established by a limited-memory quasi-Newton method. Moreover, our result identifies how memory size affects convergence speed, which provides an explicit trade-off between these two factors. To proceed, we introduce some technical conditions on the objective function $f$.

**Assumption 1.** *The function $f$ is $\mu$-strongly convex and its gradient is $L$-Lipschitz continuous.*

**Assumption 2.** *The Hessian $\nabla^2 f$ is $C_L$-Lipschitz continuous, i.e., for all $\mathbf{x}_1, \mathbf{x}_2 \in \mathbb{R}^d$, we have $\|\nabla^2 f(\mathbf{x}_1) - \nabla^2 f(\mathbf{x}_2)\| \leq C_L \|\mathbf{x}_1 - \mathbf{x}_2\|$.*

Assumptions 1 and 2 are customary in the analysis of QN methods (Nocedal & Wright, 2006)[Ch. 6]. They also together imply that $f$ is strongly self-concordant with constant $C_M = C_L/\mu^{3/2}$, i.e.,

$$\nabla^2 f(\mathbf{x}_1) - \nabla^2 f(\mathbf{x}_2) \preceq C_M \|\mathbf{x}_1 - \mathbf{x}_2\|_{\nabla^2 f(\mathbf{y})} \nabla^2 f(\mathbf{z}), \text{ for any } \mathbf{x}_1, \mathbf{x}_2, \mathbf{y}, \mathbf{z} \in \mathbb{R}^d. \tag{13}$$

With these preliminaries in place, we formally present the convergence results. Specifically, we consider the convergence criterion as the operator norm that has been commonly used in (Rodomanov & Nesterov, 2021a; Lin et al., 2021; Rodomanov & Nesterov, 2021b; Jin & Mokhtari, 2022)

$$\lambda_f(\mathbf{x}_t) := \|\nabla f(\mathbf{x}_t)\|_{\nabla^2 f(\mathbf{x}_t)^{-1}} \tag{14}$$

and show the sequence $\{\lambda_f(\mathbf{x}_t)\}_{t=1}^T$ generated by LG-BFGS can converge to zero at an explicit superlinear rate. To do so, we first state the linear convergence of $\{\lambda_f(\mathbf{x}_t)\}_{t=1}^T$ for LG-BFGS.

**Proposition 2.** *Consider LG-BFGS for problem 1 satisfying Assumptions 1-2 with $\mu$, $L$ and $C_M$. Let $\epsilon(\mu, L, C_M) = \mu^2 \ln(3/2)/(4L^2 C_M)$. Then, if $\mathbf{x}_0$ satisfies $\lambda_f(\mathbf{x}_0) \leq \epsilon(\mu, L, C_M)$, we have*

$$\lambda_f(\mathbf{x}_t) \leq \left(1 - \frac{\mu}{2L}\right)^t \lambda_f(\mathbf{x}_0). \tag{15}$$

Proposition 2 states that the weighted gradient norm $\lambda_f(\mathbf{x}_t)$ of the iterates of LG-BFGS converges to zero at least at a linear rate. This is a local linear convergence result, where the initial variable $\mathbf{x}_0$ is assumed close to the solution, i.e., $\lambda_f(\mathbf{x}_0) \leq \epsilon(\mu, L, C_M)$. Before proceeding with our superlinear analysis, we define the *relative condition number* of basis $\{\mathbf{e}_i\}_{i=1}^d$ with respect to any matrix $\mathbf{E}$.

**Definition 1.** *[Relative Condition Number] Consider a matrix $\mathbf{E}$ and a vector basis $\{\mathbf{e}_i\}_{i=1}^d$. The relative condition number of each vector $\mathbf{e}_i$ with respect to $\mathbf{E}$ is defined as*

$$\beta(\mathbf{e}_i) := \frac{\max_{1 \leq k \leq d} \mathbf{e}_k^\top \mathbf{E} \mathbf{e}_k}{\mathbf{e}_i^\top \mathbf{E} \mathbf{e}_i}. \tag{16}$$

*Moreover, we define the minimal relative condition number of the subset $\{\mathbf{e}_i\}_{i=1}^\tau$ as*

$$\beta_\tau := \beta(\{\mathbf{e}_i\}_{i=1}^\tau) = \min_{1 \leq i \leq \tau} \beta(\mathbf{e}_i).$$

The relative condition number in Definition 1 characterizes how large the linear operation $\mathbf{E}\mathbf{x}$ can change for a small change on the input $\mathbf{x}$ along the direction $\mathbf{e}_i$ for $i = 1, \ldots, d$. Indeed, based on the definition in (16), this value is larger than or equal to 1. It can be also verified that the maximum possible value for the relative condition number is $\lambda_d/\lambda_1$, where $\lambda_d$ and $\lambda_1$ are the maximal and minimal eigenvalues of $\mathbf{E}$ (we assume $\lambda_1 \leq \cdots \leq \lambda_d$). If particularizing $\{\mathbf{e}_i\}_{i=1}^d$ to the eigenvectors $\{\mathbf{v}_i\}_{i=1}^d$ of $\mathbf{E}$, i.e., $\{\mathbf{e}_i = \mathbf{v}_i\}_{i=1}^d$, the relative condition number of $\mathbf{e}_1$ with respect to $\mathbf{E}$ is equivalent to the condition number of $\mathbf{E}$, i.e., $\lambda_d/\lambda_1$, and the relative condition number of $\mathbf{e}_d$ is 1.

Moreover, $\beta(\{\mathbf{e}_i\}_{i=1}^\tau)$, abbreviated as $\beta_\tau$, measures the minimum possible relative condition number for the elements in the subset $\{\mathbf{e}_i\}_{i=1}^\tau$. When increasing the size of the set, i.e., increasing $\tau$, the minimal relative condition number decreases and approaches 1. In fact, for $\tau = d$ we obtain $\beta_d = 1$. We next use this definition to characterize the progress of Hessian approximation in LG-BFGS. We follow (Rodomanov & Nesterov, 2021a) to measure the Hessian approximation error with the trace metric $\sigma(\nabla^2 f(\mathbf{x}), \mathbf{B}) = \mathrm{Tr}(\nabla^2 f(\mathbf{x})^{-1}\mathbf{B}) - d$, which captures the distance between $\nabla^2 f(\mathbf{x})$ and $\mathbf{B}$.

**Proposition 3.** *Suppose Assumptions 1-2 hold and $\nabla^2 f(\mathbf{x}) \preceq \mathbf{B}$. Let $\mathbf{x}_+$ be the updated decision variable by equation 5, $\phi = \|\mathbf{x}_+ - \mathbf{x}\|_{\nabla^2 f(\mathbf{x})}$ the weighted update difference, $\hat{\mathbf{B}} = (1 + \phi C_M)\mathbf{B}$ the corrected Hessian approximation, $(\mathbf{s}, \mathbf{r})$ the greedily selected curvature pair by equation 6, $\mathbf{B}_+ = \mathrm{BFGS}(\hat{\mathbf{B}}, \mathbf{s}, \mathbf{r})$ the updated Hessian approximation, and $\tau$ the memory size. Then, we have*

$$\sigma(\nabla^2 f(\mathbf{x}_+), \mathbf{B}_+) \leq \left(1 - \frac{\mu}{\beta_\tau dL}\right)(1 + \phi C_M)^2 \left(\sigma(\nabla^2 f(\mathbf{x}), \mathbf{B}) + \frac{2d\phi C_M}{1 + \phi C_M}\right). \qquad (17)$$

Proposition 3 captures the contraction factor for the convergence of the Hessian approximation error in LG-BFGS, which is determined by not only the objective function but also the memory size. Specifically, the term $1 - \mu/(\beta_\tau dL)$ in equation 17 embeds the role played by the memory size $\tau$. Since $\beta_{\tau_1} \geq \beta_{\tau_2}$ for any $\tau_1 \leq \tau_2$ by Definition 1, a larger $\tau$ allows the greedy selection from a larger subset $\{\mathbf{e}_i\}_{i=1}^\tau$, yields a smaller minimal relative condition number $\beta_\tau$, and generates a smaller contraction factor that improves the update progress. While the update progress is shown in equation 17, we remark that LG-BFGS does not compute any Hessian approximation matrix $\mathbf{B}$ across iterations. By combining Propositions 2-3, we formally show the convergence of LG-BFGS.

**Theorem 1.** *Consider the setting in Proposition 2. Let $\beta_{t,\tau}$ be the minimal relative condition number of the subset $\{\mathbf{e}_u\}_{u=1}^\tau$ with respect to the error matrix $\hat{\mathbf{B}}_t - \nabla^2 f(\mathbf{x}_{t+1})$ at iteration $t$, and $t_0$ be such that $(2dL/\mu) \prod_{u=1}^{t_0}(1 - \frac{\mu}{\beta_{u,\tau}dL}) \leq 1$. Then, if $\mathbf{x}_0$ satisfies $\lambda_f(\mathbf{x}_0) \leq \mu \ln 2/(4(2d+1)L)$, we have*

$$\lambda_f(\mathbf{x}_{t+t_0+1}) \leq \prod_{u=t_0+1}^{t+t_0} \left(1 - \frac{\mu}{\beta_{u,\tau}dL}\right)^{t+t_0+1-u} \left(1 - \frac{\mu}{2L}\right)^{t_0} \lambda_f(\mathbf{x}_0). \qquad (18)$$

Theorem 1 states that the iterates of LG-BFGS converge to the optimal solution and establishes an explicit convergence rate, which depends on the linear term $(1 - \mu/(2L))^{t_0}$ and the quadratic term $\prod_{u=t_0+1}^{t+t_0}(1 - \mu/(\beta_{u,\tau}dL))^{t+t_0+1-u}$. By introducing a structural assumption on the error matrix $\hat{\mathbf{B}}_t - \nabla^2 f(\mathbf{x}_{t+1})$, we can simplifies the expression of the convergence rate and show a non-asymptotic superlinear rate in the following theorem.

**Theorem 2.** *Under the same settings as Theorem 1, suppose the minimal relative condition number $\beta_{t,\tau}$ of subset $\{\mathbf{e}_i\}_{i=1}^\tau$ or the condition number $\beta_t$ of error matrix $\hat{\mathbf{B}}_t - \nabla^2 f(\mathbf{x}_{t+1})$ is bounded by a constant $C_\beta$, i.e., either $\beta_{t,\tau} \leq C_\beta$ or $\beta_t \leq C_\beta$. Then, we have*

$$\lambda_f(\mathbf{x}_{t+t_0+1}) \leq \left(1 - \frac{\mu}{C_\beta dL}\right)^{\frac{t(t+1)}{2}} \left(1 - \frac{\mu}{2L}\right)^{t_0} \lambda_f(\mathbf{x}_0). \qquad (19)$$

Theorem 2 presents concrete takeaways demonstrating the explicit superlinear rate of LG-BFGS. For the first $t_0$ iterations, LG-BFGS converges at a linear rate that is not affected by the memory limitation. For $t \geq t_0$, once the superlinear phase is triggered, LG-BFGS converges at a superlinear rate whose contraction factor depends on the condition number $C_\beta$ of the error matrix of Hessian approximation, which in turn depends on the memory size $\tau$. This is the first result showing a superlinear rate for a fixed-size limited memory QN method, while it is important to note that it only holds for a sub-class of problems where the relative condition number of the Hessian approximation error along a low-memory subspace is well-behaved. We discuss more details in Remark 2.

**Table 1:** Comparisons between LG-BFGS and other quasi-Newton methods.

| Algorithm | Rate of Convergence | Memory | Complexity |
|---|---|---|---|
| LG-BFGS | $\min\{(1 - \frac{\mu}{L})^t, (1 - \frac{\mu}{C_\beta dL})^{\frac{t(t+1)}{2}}\}$ | $\mathcal{O}(\tau d)$ | $\mathcal{O}(\tau^2 d + \tau^4)$ |
| L-BFGS | $\gamma^t$ for some $\gamma \in (0, 1)$ | $\mathcal{O}(\tau d)$ | $\mathcal{O}(\tau d)$ |
| Greedy BFGS | $\min\{(1 - \frac{\mu}{L})^t, (1 - \frac{\mu}{dL})^{\frac{t(t+1)}{2}}\}$ | $\mathcal{O}(d^2)$ | $\mathcal{O}(d^2)$ |
| BFGS | $\min\{(1 - \frac{\mu}{L})^t, (\frac{d \ln L/\mu}{t})^{\frac{t}{2}}\}$ | $\mathcal{O}(d^2)$ | $\mathcal{O}(d^2)$ |
| BFGS with DA | Asymptotic superlinear | $\mathcal{O}(\tau d)$ $\mathcal{O}(d^2)$ | $\mathcal{O}(\tau^2 d + \tau^4)$ |

**Effect of memory size.** The fact that we only employ at most $\tau$ curvature pairs affects the convergence rate in a way previously unaddressed / unclear in the literature. Specifically, a larger $\tau$ decreases the minimal relative condition number $\beta_{u,\tau}$, reduces the contraction factor $1 - \mu/(\beta_{u,\tau} dL)$, and improves the convergence rate, but increases the storage memory and computational cost; hence, yielding an inevitable trade-off between these factors. Importantly, this result recovers the explicit superlinear rate of greedy BFGS (Rodomanov & Nesterov, 2021a) when the subset $\{\mathbf{e}_u\}_{u=1}^\tau$ increases to the entire basis $\{\mathbf{e}_u\}_{u=1}^d$, i.e., the limited memory increases to the full memory.

**Remark 2.** *There may exist pessimistic scenarios, where the assumption on the error matrix $\hat{\mathbf{B}}_t - \nabla^2 f(\mathbf{x}_{t+1})$ does not hold for a small memory size $\tau$, and the convergence rate of LG-BFGS in Theorem 1 may not be superlinear as shown in Theorem 2. However, we show in Appendix F that there exists a provable bound $C_{t,\beta}$ on the condition number $\beta_t$ of $\hat{\mathbf{B}}_t - \nabla^2 f(\mathbf{x}_{t+1})$ in any circumstances, which increases with the iteration $t$, and LG-BFGS will converge at least with a linear rate. We emphasize that this bound is the worst-case analysis established on the minimal relative condition number with a single memory $\beta_{t,1}$ (memory size one), and is only for reference.*

## 5 Discussion

In this section, we compare the convergence rates, as well as storage requirements of LG-BFGS with other quasi-Newton methods. We replace all universal constants with 1 to ease the comparisons.

**LG-BFGS.** From Theorem 2, the iterates of LG-BFGS satisfy $\frac{\lambda_f(\mathbf{x}_t)}{\lambda_f(\mathbf{x}_0)} \leq \min\{(1 - \frac{\mu}{L})^t, (1 - \frac{\mu}{C_\beta dL})^{\frac{t(t+1)}{2}}\}$. When $t < C_\beta dL \ln(dL/\mu)/\mu$, the superlinear phase is not yet triggered and the first term is smaller which implies a linear rate of $(1 - \mu/L)^t$. Once the superlinear phase is triggered, i.e., $t \geq C_\beta dL \ln(dL/\mu)/\mu$, the second term becomes smaller and iterates converge at a superlinear rate of $(1 - \mu/(C_\beta dL))^{t(t+1)/2}$. The memory storage requirement of LG-BFGS is $\mathcal{O}(\tau d)$ and its per iteration complexity is $\mathcal{O}(\tau^2 d + \tau^4)$.

**L-BFGS.** The iterates of L-BFGS converge at a linear rate (Liu & Nocedal, 1989), i.e., $|f(\mathbf{x}_t) - f(\mathbf{x}^*)| \leq \gamma^t |f(\mathbf{x}_0) - f(\mathbf{x}^*)|$ for some $\gamma \in (0, 1)$, and the storage requirement and cost per iteration of L-BFGS are of $\mathcal{O}(\tau d)$, which are comparable with the ones for LG-BFGS.

**Greedy BFGS.** Based on (Rodomanov & Nesterov, 2021a; Lin et al., 2021), the iterates of Greedy BFGS satisfy $\frac{\lambda_f(\mathbf{x}_t)}{\lambda_f(\mathbf{x}_0)} \leq \min\{(1 - \frac{\mu}{L})^t, (1 - \frac{\mu}{dL})^{\frac{t(t+1)}{2}}\}$. It requires $dL \ln(dL/\mu)/\mu$ iterations to trigger the superlinear phase and achieves a faster superlinear rate than LG-BFGS since $C_\beta \geq 1$. However, the storage and per iteration complexity of greedy BFGS are $\mathcal{O}(d^2)$.

**BFGS.** The iterates of BFGS satisfy $\frac{\lambda_f(\mathbf{x}_t)}{\lambda_f(\mathbf{x}_0)} \leq \min\{(1 - \frac{\mu}{L})^t, (\frac{d \ln(L/\mu)}{t})^{\frac{t}{2}}\}$ (Rodomanov & Nesterov, 2021b). The superlinear convergence of BFGS starts after $d \ln(L/\mu)$ iterations, while its superlinear rate is slower than LG-BFGS, i.e., $(1 - \frac{\mu}{C_\beta dL})^{\frac{t(t+1)}{2}} \ll (\frac{d \ln \frac{L}{\mu}}{t})^{\frac{t}{2}}$. Moreover, the storage requirement of LG-BFGS and its cost per iteration are smaller than $\mathcal{O}(d^2)$ of BFGS.

**BFGS with displacement aggregation** only guarantees an *asymptotic* superlinear convergence (Sahu & Pattanaik, 2023b; Berahas et al., 2022a). Moreover, it is not strictly limited-memory as it computes the variable variation as $\mathbf{s}_t = \mathbf{x}_{t+1} - \mathbf{x}_t$ which can be any vector in $\mathbb{R}^d$. Hence, $\mathbf{s}_t$ could be independent of all previously stored variable variations, which leads to (C1) in the displacement step and increases the memory size [cf. equation 7]. As a result, the memory size could increase up to $d$, resulting in a storage and cost per iteration of $\mathcal{O}(d^2)$. Table 1 summarizes the comparisons.

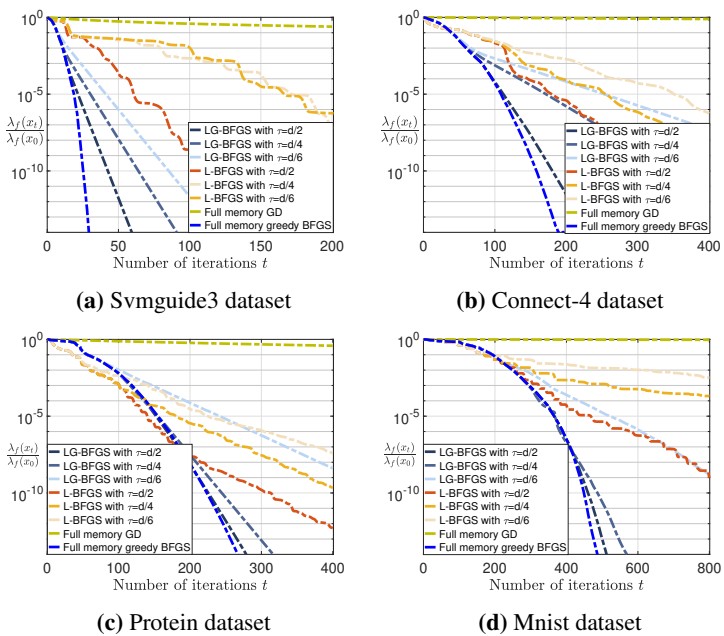

**Figure 1:** Comparison of LG-BFGS, L-BFGS, gradient descent, and greedy BFGS on four datasets.

## 6 EXPERIMENTS

We compare the performance of LG-BFGS with gradient descent, L-BFGS, and greedy BFGS on different datasets. We focus on the logistic regression with $l_2$ regularization where $f(\mathbf{x}) = \frac{1}{N}\sum_{i=1}^{N}\ln(1+\exp(-y_i\mathbf{z}_i^\top\mathbf{x}))+\frac{\mu}{2}\|\mathbf{x}\|^2$. Here, $\{\mathbf{z}_i\}_{i=1}^{N}\in\mathbb{R}^d$ are data points and $\{y_i\}_{i=1}^{N}\in\{-1,1\}$ are labels. We normalize all samples s.t. the objective function gradient is smooth with $L = 1/4+\mu$. Considering the local nature of superlinear results for QN methods, we construct a setup with a warm start, i.e., the initialization is close to the solution. Details may be found in Appendix G, where we also analyze performance with a cold start. We run experiments on four datasets: svmguide3, connect-4, protein and mnist, and select the regularization parameter $\mu$ to achieve the best performance, whose descriptions are also in Appendix G. We set the stepsize of all QN methods as 1, while the stepsize of gradient descent is $1/L$. The subset $\{\mathbf{e}_i\}_{i=1}^{\tau}$ is selected: (i) when the number of stored curvature pairs is smaller than $\tau$, we select $\mathbf{s}_t$ from the entire basis $\{\mathbf{e}_i\}_{i=1}^{d}$; (ii) when the number of stored curvature pairs reaches $\tau$, we select $\mathbf{s}_t$ from the stored variable variations $\{\mathbf{s}_u\}_{u=0}^{\tau-1}$.

In Figure 1, we observe that LG-BFGS consistently outperforms L-BFGS and gradient descent, corroborating our theoretical results. The convergence rate of LG-BFGS degrades with the decrease of memory size as expected. This is because a smaller memory restricts the selection space of the subset $\{\mathbf{e}_i\}_{i=1}^{\tau}$ in equation 6, which increases the minimal relative condition number $\beta_\tau$ in equation 17 and decreases the convergence of the Hessian approximation matrix in equation 17. Hence, it results in a slower superlinear rate of LG-BFGS and the latter may be close to a linear rate when the memory size is small. Moreover, LG-BFGS can exhibit a comparable performance to greedy BFGS when the memory size is large. This corresponds to the fact that Theorem 1 recovers the superlinear rate of greedy BFGS when the memory size is the space dimension, i.e., $\tau = d$.

## 7 CONCLUSION

In this work, we developed the LG-BFGS method as an innovative synthesis of greedy basis vector selection and displacement aggregation. The resultant method attains the first non-asymptotic superlinear guarantee for limited-memory quasi-Newton methods, which also explores an explicit effect of the memory size on the convergence rate. We consider this work as the first stride to provably show the possibility of achieving superlinear rates with memory limitation, which lays a foundation to develop extended limited memory methods offering more refined trade-offs between rate and cost.

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
