# OpenReview forum: "Limited-Memory Greedy Quasi-Newton Method with Non-asymptotic Superlinear Convergence Rate"
_ICLR.cc/2024/Conference — Submitted to ICLR 2024_

### Official Review · Reviewer_hAw3 · 2023-10-25

**Soundness:** 2 fair
**Presentation:** 2 fair
**Contribution:** 2 fair
**Rating:** 5
**Confidence:** 4

**Summary:**

This paper proposed a variant of limited-memory BFGS method by incorporating the techniques of greedy updates (Rodomanov and Nesterov 2021a) and the strategy of dynamically selecting the curvature pairs (Berahas 2022a). The convergence analysis show that the proposed methods can achieve explicit local superlinear rates.

**Strengths:**

This paper studies a historical open problem for limited memory quasi-Newton methods: can limited mememory quasi-Newton methods achieve explicit local superlinear rates or better linear rates than the first-order methods. The incorporation of selecting curvature pairs and the greedy quasi-Newton methods is interesting. The results could be super exiciting to the community if they are correct.

**Weaknesses:**

The authors define condition number on the error matrix $\hat{B}\_t-\nabla^2 f(x\_{t+1})$ and suppose it can be bounded. This is a very strong and impractical assumption. During the optimizing process, some eigenvalues of $\hat{B}\_t-\nabla^2 f(x\_{t+1})$ could be $0$, which makes the condition number unable to be bounded (i.e. for the greedy quasi-Newton methods (Rodomanov and Nesterov 2021a), which is the full memory version of the proposed methods, we have $\hat{B}_t\to\nabla^2 f(x^*)$ ).

The author provide the bound on the condition $\beta_t$ in Appendix F, which results an linear rate instead of the superlinear rate, however $e^{-Ct}\approx (1-C)^t$ where $C=q^{t_0+1}\mu/(C_\beta dL)\ll 1/\kappa$. Such rate is even worse than the linear rate of gradient descent which cannot be claimed as an "improved linear rate''.

Given upon this, I think "close the gap of showing non-asymptotic superlinear rate of limited memory quasi-Newton methods'' in the abstract is overclaimed.

**Questions:**

Please refer to the weakness part.

---

> ### Author Response · Authors · 2023-11-20
> **Response by Authors**
>
> We would like to thank the Reviewer for their time and effort spent in this manuscript and for the constructive feedback which helped improve the paper. In the sequel, we provide our response and actions to each of the Reviewer's comments.
>
> **W1**: Thank you for the comment. We would like to clarify that the goal of this paper is to explore the possibility of achieving an explicit superlinear rate with limited memory and characterize the role of the memory size in the convergence rate. On the one hand, we provide a definitive answer to this question, i.e., we provably show that it is possible to achieve non-asymptotic superlinear rates for limited-memory methods and analyze how the memory size affects the convergence rate explicitly via the minimal relative condition number. On the other hand, we identify the sufficient condition required for achieving this superlinear rate, i.e., the minimal relative condition number $\beta_{\tau}$ of the error matrix is bounded. Both facts have not been studied so far.
>
> We agree with the Reviewer that there may exist pessimistic scenarios where the boundness assumption may not hold and the convergence rate of the LG-BFGS in Theorem 1 may reduce to be linear. This indicates that while it is possible to achieve superlinear rates, it requires certain conditions, which identifies the inevitable limitation brought by the memory shortage. However, even in the worst case, the convergence rate of the LG-BFGS is at least linear as stated in Proposition 1. These results provide a clear theoretical explanation for the relationship between the convergence rate and the memory limitation. We will clarify this point and include the above discussions in the final upload.
>
> **W2**: We agree with the Reviewer that the bound provided in Appendix F is not ideal but only for reference, and the rate derived from this bound is not superlinear. However, we would like to clarify that this bound is the worst-case analysis established on the condition number, i.e., the minimal relative condition number with a single memory $\beta_{1}$ (memory size one). Moreover, even in the worst scenario without bound, the LG-BFGS retains the linear convergence rate as stated in Theorem 1 and outperforms the gradient descent as corroborated in our experiments. We will clarify this point in the final upload. Thank you for the comment.
>
> **W3**: Thank you for the comment. We would like to clarify that the contributions of this work are to answer three questions that have been unclear in the literature: (i) Is it possible to achieve superlinear rates for limited-memory methods? (ii) If so, what conditions are required to make it possible? (iii) How does the memory size affect the convergence rate? This paper provides a definitive answer to the first question, identifies the exact condition / assumption required for superlinear convergence to the second question, and furnishes a clear expression for the non-asymptotic convergence of the proposed LG-BFGS that delineates the direct influence of memory size through the concept of relative condition number to the third question.
>
> Moreover, we highlight that our assumption and results are consistent with the full-memory BFGS method, i.e., the bound of $\beta_\tau$ reduces to $1$ and the superlinear rate becomes that of the full-memory method when the memory size $\tau$ increases to the feature dimension $d$. This builds a clear path from the limited-memory method to its full-memory counterpart, and provides an explicit theoretical explanation behind their performance changes along the reduction of memory size. All these aspects have not been studied so far and thus, we consider this work as the first strade that closes the gap and believe these are novel contributions for the community to further explore the limited-memory methods.
>
> Finally, we remark that even with the boundness assumption, providing an explicit non-asymptotic superlinear rate for limited-memory BFGS methods is already a challenging and tedious task. This requires: (i) developing a novel combination of greedy selection and displacement aggregation to maximize the update progress within the limited-memory framework; (ii) defining a novel concept that embeds the impact of memory limitation in the convergence characterization; (iii) providing novel theoretical derivations that are capable of leveraging the advantages of both techniques to achieve non-asymptotic superlinear convergence with memory limitation.
>
> We will be sure to appropriately caveat the abstract and introduction to clarify the restricted setting in which our results hold, so as to ensure we are not overclaiming. Thank you again for your insightful comment.

---

> > ### Comment · Reviewer_hAw3 · 2023-11-23
> >
> > Thanks for your response.  I understand that this paper makes some contribution to the theory of limited-memory quasi-Newton methods and decide to raise my score from 3 to 5.
> >
> > However, the theory obtained in this paper is limited. The bound of $\beta_{t,\tau}$ or $\beta_t$ does not rely on the object function itself only, but also heavily rely on the procedure of the algorithm ($B_t$ is generated by the algorithm). I suggest the author to find some simple class of function that fulfill the assumption of bounded relative condition number of the error matrix.

---

> > > ### Author Response · Authors · 2023-11-23
> > > **Response by Authors**
> > >
> > > We wish to thank the Reviewer for the consideration on our responses, and we sincerely appreciate the Reviewer's willingness to reconsider the initial score.
> > >
> > > Thank you for the additional feedback. We would like to clarify that this paper is the first stride towards the superlinear convergence of limited-memory BFGS methods in the literature, the goal of which is to explore the possibility of achieving a non-asymptotic superlinear convergence with limited memory and to provide the explanation behind the inner-working mechanism of limited-memory BFGS methods. Specifically, we (i) develop a limited-memory algorithm that is possible to achieve an explicit superlinear rate; (ii) identify the condition under which this could be possible; (iii) characterize the effect of the memory size on the convergence rate. With these endeavors in mind, we believe this work lays a foundation for the community to develop extended limited-memory methods and generalize the convergence analysis in future research.
> > >
> > > On the other hand, we agree with the Reviewer that interesting topics for future research include finding specific classes of objective functions that fulfill the condition and guarantee the superlinear rate in any circumstances. We will clarify this point in the final upload. Thank you again for the suggestion.

---

### Official Review · Reviewer_AdGH · 2023-10-30

**Soundness:** 3 good
**Presentation:** 4 excellent
**Contribution:** 3 good
**Rating:** 5
**Confidence:** 4

**Summary:**

This paper proposes and analyzes LG-BFGS, a greedy version of the celebrated
L-BFGS quasi-Newton method. The modifications are two-fold: (i) greedy selection
of the parameter difference vector $s_t$ from a truncated basis, and (ii) a
so-called _displacement step_ which re-wights the curvature pair history to
capture new information rather than simply replacing the oldest pair with the
new one. The authors leverage these modifications to prove a local super-linear
convergence rate for LG-BFGS; this rate is particularly nice in that it
improves with the size of the history.
The paper concludes with experimental comparison of LG-BFGS, Greedy BFGS,
and other related methods.

**Strengths:**

The major strength of this paper is the convergence analysis for LG-BFGS,
which shows a super-linear rate and is sensitive enough to improve with the size
of the history $\tau$. This is a strong achievement given the long history
of interesting in limited-memory quasi-Newton methods.

Other notable strengths include:

- The LG-BFGS method represents a novel synthesis of ideas from the
    quasi-Newton literature, combining greedy basis selection with careful
    updates to the history.

- LG-BFGS has fast per-iteration convergence (locally) in practice,
    particularly when compared with the standard L-BFGS method.

- The text is polished and contains very few typos.

Note that I did not check the proofs for correctness.

**Weaknesses:**

I have several concerns with this work.

- The version of LG-BFGS which has super-linear convergence uses a correction
    strategy that requires Hessian-vector products. The naive cost of these
    computations are $O(d^2)$, which would make LG-BFGS more expensive than
    Greedy BFGS in practice.  However, this computational cost is not addressed
    anywhere in the text.

- Related to the previous point, the authors repeatedly miss-represent the
    computational complexity of LG-BFGS as being comparable to L-BFGS, which it
    is formally not.

- LG-BFGS even without the correction factor has a significant per-iteration
    time cost and experiments in the appendix show that L-BFGS or G-BFGS
    may be preferable in practice instead.

- Moreover, the role of the correction strategy is not studied empirically
    despite the necessity of this trick for deriving convergence guarantees.
    Experimental results appear to be shown only for the algorithm without
    correction, which does not match the analysis in the paper.

Given my concerns, I feel the submission is borderline. However, I am willing
and would like to update my score if the authors can address these issues as
well as my questions below.

**Questions:**

- Curvature Pair Update: It seems $e_i$ stands for both a general basis and
    for the standard basis. This overloading of notation is awkward since $e_i$
    is typically a standard basis vector and becomes confusing in
    Definition 1, where it's not obvious if $e_i$ now refers to the standard
    or general basis I suggest introducing a separate notation for the general
    basis and using $e_i$ only for the standard one.

- Remark 1: The computational cost is of a different order for L-BFGS and LG-BFGS:
    $O(\tau^2 d + \tau^4)$ vs $O(4 \tau d)$, so I would not say that they are comparable.
    If $\tau \leq 4$ is very small, then the are comparable, but not in an
    asymptotic sense.  Moreover, if you are going to use big-O notation, then
    the complexity of L-BFGS should be written as $O(\tau d)$ since the
    constant $4$ does not contribute to the asymptotic growth.
    The first comment also applies to the discussion in Section 5.

- Section 3.1: Computing $\phi_t$ requires a Hessian-vector product, right?
    This should be $O(d^2)$, since $x_{t+1} - x_t$ doesn't have any special structure,
    unlike the Hessian-vector products with the variable variation $s_t$.
    Why doesn't this contribute to the overall complexity of LG-BFGS as stated
    in Section 5?

- Proposition 3: What matrix is the minimal relative condition number defined
    with respect? Is it the same error matrix from Theorem 1?
    I don't see that stated anywhere, if so.

- Equation 17: Under what conditions is this actually a contraction? My
    understanding is that $\tau < d$ implies the trace progress condition
    cannot converge to $d$ unless the Hessian is low-rank and spanned by $e_1,
    \ldots, e_\tau$. It would be nice to see some discussion of this fact.

- Comparison to BFGS: Do you think the slower rate of BFGS compared to G-BFGS
    is an artifact of the analysis by Rodomanov and Nesterov, or is it because
    of the different update strategy used for the curvature matrix?
    The per-iteration convergence of L-BFGS and LG-BFGS suggets it is the latter.

- Experimental Comparison: It doesn't make sense to compare G-FBGS to LG-BFGS
    with any choice of $\tau$ as a linear function of $d$. In this setting,
    LG-BFGS is asymptotically more expensive than G-BFGS and both thereotically and
    experimentally slower than G-BFGS. It only makes sense to use LG-BFGS when
    $\tau$ is an absolute constant or a slowly growing sub-linear function of
    $d$. For example, choosing $\tau \in \{5, 10, 25, 50\}$ would be
    appropriate for MNIST and Protein, while somewhat smaller choices would be
    suitable for Connect-4.

- Appendix G:

    - Do the experiments in the main paper ignore the correction strategy
    and set $\tau r_t = r_t$, or this is only done in the appendix experiments?
    This seems important as computing the Hessian-vector product needed for the
    correction is computationally expensive, but also apparently necessary
    for a theoretical convergence guarantee. I think it is important to
    provide an ablation study comparing the performance of LG-BFGS with and
    without the correction factor (in wall-clock time) so that its effects
    can be properly understood.

    - L-BFGS is much more competitive with LG-BFGS when convergence is shown
    in terms of wall-clock time (Figure 2 in the appendix). Indeed, even
    G-BFGS becomes competitive with LG-BFGS when measured in wall-clock time.
    This worries me, since (i) the results are shown without the
    potentially expensive correction step needed for convergence guarantees;
    and (ii) L-BFGS is much simpler than LG-BFGS, so the relative merits of
    LG-BFGS are somewhat diminished.

---

> ### Author Response · Authors · 2023-11-20
> **Response by Authors**
>
> We would like to thank the Reviewer for spending time thoroughly reviewing the manuscript and providing constructive feedback. We are glad to see the Reviewer appreciated the novelty and the presentation of the work. In the sequel, we provide our response and actions to each of the Reviewer's comments and questions.
>
> **W1, W2, Q3**: We agree with the Reviewer that the weighted norm $\phi_t=\lVert x_{t+1}-x_t\rVert_{\nabla^2f(x_t)}$ in the correction strategy requires the computation $O(d^2)$. However, we would like to clarify that this weighted norm can be replaced by $\hat{\phi_t}=\lVert x_{t+1}-x_{t}\rVert_{LI}$, which provides the same correction guarantees and only requires the computation $\mathcal{O}(d)$.
>
> Specifically, from $\mu I\preceq\nabla^2 f(x_t)\preceq LI$ of Assumption 1, we have $\hat{\phi}_t\ge\phi_t$ and $(1+\hat{\phi}_tC_M)B_t\succeq (1+\phi_tC_M)B_t\succeq B_t$, and thus $\hat{\phi}_t$ provides the same correction guarantees as $\phi_t$. Since $LI$ is a diagonal matrix, $\hat{\phi}_t$ only requires the computation $O(d)$. Additionally, we note that this modification has an impact. It leads to a larger error between the Hessian approximation $\hat{B}_t$ and the actual Hessian $\nabla^2f(x_t)$, subsequently reducing the update progress of algorithm. We will clarify this point and include this modification in the final upload. Thank you for pointing this out.
>
> **W3, Q8.2**: Thank you for the comment. We would like to clarify this point from two aspects:
>
> (i) While the proposed LG-BFGS requires more worst-case computation than L-BFGS, it achieves a faster convergence rate w.r.t. iteration in all experiments -- see Fig. 1 in Section 6. In the experiments of the appendix, we see that LG-BFGS is faster w.r.t. runtime than L-BFGS in scenarios (a), (b) and (d), while we concede that it may be slower in some scenarios like (c) (where these scenarios are for specific relationships on memory parameter $\tau$ and input dimension $d$). This indicates an inevitable trade-off between the convergence rate and the computational cost.
>
> (ii) We propose LG-BFGS as the first step to explore the possibility of achieving a non-asymptotic superlinear convergence with limited memory and analyze the explicit effect of memory limitation on the convergence rate. We are currently working on potential extensions to further reduce the worst-case computation of LG-BFGS. For example, we may develop a method that harmoniously combines LG-BFGS and L-BFGS in a compatible way akin to [a], thereby leveraging the strengths from both sides. With these endeavors in mind, we believe this work lays a foundation to develop extended limited-memory methods, offering more refined trade-offs between rate and cost in future research.
>
> [a] Sharpened Quasi-Newton Methods: Faster Superlinear Rate and Larger Local Convergence Neighborhood.
>
> **W4, Q8.1**: Thank you for the comment. We would like to clarify that we do not apply the correction strategy in all experiments, following the existing greedy BFGS methods (Rodomanov & Nesterov 2021a; Lin et al. 2021). Specifically, the application of the correction strategy is to guarantee the error matrix positive semi-definite, i.e., $\hat{B_t}-\nabla^2f(x_{t+1})\succeq 0$. However, in practical implementation, the error matrix is positive definite without correction and thus, there is no need to apply the correction strategy, as recommended by (Rodomanov & Nesterov 2021a; Lin et al. 2021). To further address the Reviewer's concern, we will provide an ablation study comparing the performance of LG-BFGS with and without the correction factor in the final upload.
>
> **Q1**: We agree with the Reviewer that using this notation for both bases is ambiguous. We will disambiguate the notation of general versus standard basis vectors so the algorithm executation is clear. Specifically, we will use $g_i$ to represent a general basis and $e_i$ to represent the standard one in the revised paper. Thank you for the suggestion.
>
> **Q2**: We agree with the Reviewer and we will endeavor to be consistent with the use of complexity notation, to omit constant factors that are not problem-dependent. We thank the reviewer for helping us clarify the algorithm's required computational effort.
>
> **Q4**: The Reviewer is correct that it is the error matrix $\hat{B_t}-\nabla^2f(x_{t+1})$ in Theorem 1, and we will explicitly state this fact. Thank you for pointing this out.

---

> > ### Comment · Reviewer_AdGH · 2023-11-20
> >
> > **W1, W2, Q3**: Right, this makes sense. I think it's important that this modification is made very clear in the paper if the authors want to continue to emphasize the $O(\tau^2 d + \tau^4)$ (i.e. sub-quadratic in $d$) computational cost of LG-BFGS. My guess is that Proposition 2 doesn't change too much, but I assume $\phi$ will be replaced by $\hat \phi$ in Proposition 3 --- is this correct?
> >
> > It's much less clear to me what the effects on Theorem 1 will be. It seems like the minimal relative condition number will change since the error matrix will have much larger eigenvalues. Is this the correct interpretation and, if so, can you please comment on the ramifications for the super-linear convergence rate?
> >
> > **Other Questions**: I'm satisfied by the other responses. Thanks!

---

> ### Author Response · Authors · 2023-11-20
> **Response by Authors**
>
> (cont.)....
>
> **Q5**: The Reviewer raises a good point. We would like to clarify that assuming $\beta_{t,\tau}$ is bounded by $C_\beta$, the contraction factor $(1-\frac{\mu}{\beta_{t,\tau}dL})$ in (17) is bounded by a constant, i.e., $(1-\frac{\mu}{\beta_{t, \tau}dL}) \le (1-C)$, while $\phi_t\le\lambda_f(x_t)$ decreases to zero at a linear rate from Theorem 1. Thus, (17) would become a contraction when the iteration is large or the initialization is good. Moreover, the memory size $\tau$ determines the value of the contraction factor, i.e., a larger $\tau$ yields a smaller $\beta_{t, \tau}$, leads to a smaller contraction factor, increases the convergence rate. We will add more discussions regarding this fact in the final upload. Thank you for the suggestion.
>
> **Q6**: We agree with the Reviewer that the rate improvement of Greedy BFGS compared to BFGS is because of the greedy update strategy used for the curvature matrix. Specifically, the greedy update strategy of G-BFGS selects the variable variation $s_t$ as the basis vector that maximizes the weighted norm ratio [cf. (6)]. It optimizes the update progress of the Hessian approximation [cf. (41) in the proof of Proposition 2] and improves the convergence rate. We will clarify this point in the revised paper. Thank you for the comment.
>
> **Q7**: Thank you for the comment. If $\tau$ is small, the performance of LG-BFGS will degrade close to L-BFGS, though the computation of LG-BFGS decreases. The reason we explored this range of $\tau$ is to illuminate the performance tradeoffs between these extremes. In general, we agree selecting $\tau$ as an absolute constant makes sense, but doing so would not reveal how performance varies with respect to the interplay between memory size and feature dimension. We will clarify this selection of memory parameter and conduct this additional numerical study in the final upload, if afforded the opportunity.

---

> ### Author Response · Authors · 2023-11-21
> **Response by Authors**
>
> We wish to thank the Reviewer for the consideration on the answers and for the additional feedback. We are glad to see the Reviewer found our answers helpful.
>
> **W1, W2, Q3**: The Reviewer is correct that this modification on the correction strategy has an impact on the convergence results. Specifically, from $\mu I\preceq\nabla^2f(x_t)\preceq LI$, we have $\phi_t\le\hat{\phi}_t\le\frac{L}{\mu}\phi_t$ and $\phi_tC_M\le\hat{\phi}_tC_M\le\phi_t\frac{L}{\mu}C_M$ in the correction strategy. This indicates that while this modification may increase the error between the Hessian approximation $\hat{B_t}$ and the actual Hessian $\nabla^2 f(x_t)$, this increase is limited. For Proposition 2, it will reduce the update progress in (24) of the proof, necessitating a slightly better initialization to establish a bound in (27) to derive the linear convergence. For Proposition 3, $\phi_t$ will be replaced by $\hat{\phi}_t$ as the Reviewer points out. For Theorem 1, it will reduce the update progress in (59) and (60) of the proof, and need a slightly better initialization to establish a bound from (64) to (65) for deriving the superlinear convergence. Additionally, the Reviewer is correct that the minimal relative condition number will be w.r.t. the modified error matrix, which may have slightly larger eigenvalues. We will include these modifications and clarify these details clearly in the final upload. Thank you again for pointing this out.

---

> > ### Author Response · Authors · 2023-11-23
> > **Response by Authors**
> >
> > Dear Reviewer,
> >
> > We sincerely appreciate your constructive comments and suggestions that strength our paper. If our responses have addressed your concerns, we humbly hope that the Reviewer could revisit the score in light of the responses and revisions.
> >
> > Thank you again for your time and effort in reviewing our work.

---

### Official Review · Reviewer_XXyA · 2023-10-31

**Soundness:** 2 fair
**Presentation:** 2 fair
**Contribution:** 1 poor
**Rating:** 3
**Confidence:** 3

**Summary:**

In this work, the authors focus on the non-asymptotic convergence analysis of quasi-Newton optimization methods. This study addresses the challenge of balancing computational complexity and memory requirements in such methods. While prior approaches demonstrated a local superlinear rate, they suffered from high memory demands due to the storage of past curvature information. Limited-memory variants, like the L-BFGS method, reduced these demands by utilizing a limited window of curvature information. However, prior to this work, there was no known limited-memory quasi-Newton method that could achieve non-asymptotic superlinear convergence. The authors introduce the Limited-memory Greedy BFGS (LG-BFGS) method, which incorporates techniques like displacement aggregation and basis vector selection to balance memory requirements while achieving a superlinear rate of convergence. This work reveals an explicit trade-off between convergence speed and memory usage, a novel contribution to the field. Numerical experiments support their theoretical findings, confirming the method's effectiveness.

**Strengths:**

* In this paper the authors provide a nonasymptotic local superlinear rate for the LG-BFGS method with affordable storage requirements.
* They establish an explicit trade-off between the memory size and the contraction factor that appears in the superlinear rate.
* The LG-BFGS method uses greedy basis vector selection for the variable variation and displacement aggregation on the
gradient variation.
* The authors provide an experimental comparison of the proposed method with gradient descent, L-BFGS, and greedy BFGS.
* From experiments one can observe that the performance of the proposed algorithm is comparable with the greedy BFGS when the memory size is large.

**Weaknesses:**

* It would be advisable to present the contributions as bullet points to provide a clearer view.
* In Figure 1(a) the second LG-BFGS with $\tau = d/6$ is a lapsus right? Moreover, it will be helpful to change the color of the Full memory GD as it gets confused with LG-BFGS with $\tau = d/2$.
* It would be beneficial to include a table outlining the convergence rate and computation complexity of the proposed method and other non-asymptotic methods.
* There is missing citations and work comparison with [1].
* The title of the paper may lead one to believe that the proposed method is applicable to Quasi-Newton methods, including SR1. However, upon reading the paper, it becomes clear that the authors only discuss BFGS. To avoid any ambiguity, it would be better to modify the title accordingly.
* The paper presents only theoretical proofs about the local convergence of the proposed method. However, it lacks discussion on how the method can achieve global convergence. On the other hand, in [2], the authors demonstrate both global and local convergence rates that make their method superior to the proposed one.

[1] Sahu, Manish Kumar, and Suvendu Ranjan Pattanaik. "Non-asymptotic superlinear convergence of Nesterov accelerated BFGS."

[2] Jiang, Ruichen, Qiujiang Jin, and Aryan Mokhtari. "Online Learning Guided Curvature Approximation: A Quasi-Newton Method with Global Non-Asymptotic Superlinear Convergence." arXiv preprint arXiv:2302.08580 (2023).

**Questions:**

Mentioned above.

---

> ### Author Response · Authors · 2023-11-20
> **Response by Authors**
>
> We would like to thank the Reviewer for their time spent in this paper and for the insightful comments which helped improve the paper. We are glad to see the Reviewer found the contribution novel. In what follows, we provide our response and actions to your concerns.
>
> **W1**: Thank you for the suggestion. We will revise the presentation of our contributions in the final upload to be itemized bullets as follows:
>
> 1. We propose a Limited-memory Greedy (LG)-BFGS method, which is a novel synthesis of greedy basis selection and displacement aggregation. It maximizes the update progress per iteration to improve the convergence, while maintaining the limited memory setting.
>
> 2. We furnish a clear and explicit expression for the non-asymptotic convergence of the LG-BFGS, which not only delineates the direct influence of memory size through the concept of relative condition number $\beta_\tau$ [Def. 1], but also identifies roles played by other factors (e.g., problem properties, initial conditions, etc.). This is the first work that characterizes the explicit effect of memory size on the convergence rate.
>
> 3. The LG-BFGS presents an non-asymptotic superlinear rate with the boundness assumption on $\beta_\tau$. This is the first work that shows the possibility of achieving superlinear rates in limited-memory methods and identifies the exact condition / assumption under which this could be possible. The result recovers that of the full-memory method when the memory size increases to the feature dimension. This establishes a clear path from limited-memory methods to full-memory ones, and provides an explicit theoretical explanation behind the performance change along the reduction of memory size
>
> In summary, this paper answers three unknown questions: (i) Is it possible for limited-memory methods to achieve explicit superlinear rates? (ii) If so, which condition does it require? (iii) What is the explicit effect of memory size on the convergence rate?
>
> **W2**: Thank you for the suggestion. We have changed the color, which will be reflected in the final version of this work.
>
> **W3**: Thank you for the suggestion. We will include the table to summarize the comparision of convergence rate and computation complexity in Section 5, which will be comparable to the one below:
>
> | Algorithm | Rate of Convergence | Memory | Complexity |
> | -------- | -------- | -------- | -------- |
> | LG-BFGS (This work) | $\min\{(1-\frac{\mu}{L})^t, (1-\frac{\mu}{C_\beta dL})^{\frac{t(t+1)}{2}}\}$ | $\mathcal{O}(\tau d)$ | $\mathcal{O}(\tau^2 d+\tau^4)$ |
> | L-BFGS | $\gamma^t$ for some $\gamma \in (0, 1)$ | $\mathcal{O}(\tau d)$ | $\mathcal{O}(\tau d)$ |
> | Greedy BFGS | $\min\{(1-\frac{\mu}{L})^t, (1-\frac{\mu}{dL})^{\frac{t(t+1)}{2}}\}$ | $\mathcal{O}(d^2)$ | $\mathcal{O}(d^2)$ |
> | BFGS | $\min\{(1-\frac{\mu}{L})^t, (\frac{d \ln{L/\mu}}{t})^{\frac{t}{2}}\}$ | $\mathcal{O}(d^2)$ | $\mathcal{O}(d^2)$ |
> | BFGS with DA | Asymptotic superlinear | $\mathcal{O}(\tau d)$~$\mathcal{O}(d^2)$ | $\mathcal{O}(\tau^2 d + \tau^4)$ |
>
> **W4**: Thank you for the suggestion. We will add this citation, and are grateful for the Reviewer to bringing this innovative work to our attention. We have reproduced the manner in which it will be discussed in the paper here for convenience:
>
> Accelerations of standard BFGS have been considered in various forms; however, Nesterov acceleration is one approach that has been shown to enhance the rate of convergence of standard BFGS after a threshold number of iterations, under suitable choice of step-size and error-bound tolerance [1]. However, we note that, similar to standard BFGS, accelerated variants require full memory to achieve these rates.
>
> **W5**: We agree with the Reviewer that the title may give the impresion that our results hold for the entire Broyden class of Quasi-Newton methods, or SR1, whereas our developments are only for BFGS. However, similar techniques could be applied to these alternative approaches. We will be more careful to caveat these points of distinction in the abstract and introduction so as to clarify the interpretation. Thank you for the suggestion.
>
> **W6**: We agree with the Reviewer that [2] provides an analysis methodology for both local and global convergence. However, we would like to clarify that this work requires full memory for implementation, i.e., it requires the storage of the previous Hessian approximation matrix or instead storing all past curvature information, which is different from our limited-memory setting. Moreover, it was contemporaneous of ours, and marks a significant departure from a long history of analyzing the local rate of convergence of Quasi-Newton methods. It is a valid scope of future work to explore how explicit superlinear rates in the limited-memory setting may be achieved globally. We will add this work and include the above discussion in the final upload. Thank you for pointing this out.

---

> > ### Comment · Reviewer_XXyA · 2023-11-21
> >
> > Dear Authors,
> >
> > Thank you for your reply. However, I have not observed any revisions to the manuscript that align with the information you provided.

---

> ### Author Response · Authors · 2023-11-21
> **Response by Authors**
>
> We wish to thank the Reviewer for the consideration on our responses and for the additional feedback.
>
> We have updated the revised manuscript in the system based on our responses to your comments. Thank you again for providing high-quality comments and nice suggestions, which enabled us to strengthen our paper.

---

> > ### Comment · Reviewer_AdGH · 2023-11-22
> >
> > Has the updated manuscript been uploaded to OpenReview? The system says that there are no revisions to display and the PDF seems to be the same.
> >
> > I would like to see the updated discussion around the correction factor and the impact it has on the convergence rate.

---

> ### Author Response · Authors · 2023-11-22
> **Response by Authors**
>
> Thank you for the feedback regarding the update issue of the system. We have re-updated the manuscript in the system and please let us know if you still cannot see the revisions in the updated manuscript.
>
> Specifically, we have made the following revisions based on your suggestions:
>
> (i) We have revised the second paragraph of Section 1 to include the references suggested by the Reviewer;
>
> (ii) We have revised the proposed research questions and the contributions in Section 1, to itemize and highlight our contributions;
>
> (iii) We have added the sentence after the contributions in Section 1, to clarify that "our development focuses on the BFGS method in the class of quasi-Newton methods, while similar techniques could be applied to its alternative variants".
>
> (iv) We have added Table I to summarize the comparisons in Section 5.
>
> (v) We have changed the color of GD and fixed the lapsus in Figure 1 in Section 6 to avoid any confusion.
>
> (vi) We have revised the definition of $\phi_t$ in the correction strategy and clarified that "it only requires the computation $O(d)$ because $LI$ is a diagonal matrix" in the first paragraph of Section 3.1. Moreover, we have revised the initial condition in Proposition 2 and Theorem 1 to update the impact of this modification (Theorem 1 considers the same settings as Proposition 2 and thus, there is no change in Theorem 1). Additionally, since the correction factor $\phi_t$ has been updated with the modified one, we do not need to change Proposition 3 and the contraction factor in Theorem 1.
>
> Thank you again for the constructive suggestions and the additional feedback. Please let us know if there is still any problem of seeing the revisions in the updated manuscript of the system.

---

> > ### Author Response · Authors · 2023-11-23
> > **Response by Authors**
> >
> > Dear Reviewer,
> >
> > We sincerely appreciate your constructive comments and suggestions that strength our paper. If our responses have addressed your concerns, we humbly hope that the Reviewer could revisit the score in light of the responses and revisions.
> >
> > Thank you again for your time and effort in reviewing our work.

---

### Official Review · Reviewer_PF4F · 2023-11-06

**Soundness:** 3 good
**Presentation:** 3 good
**Contribution:** 3 good
**Rating:** 6
**Confidence:** 2

**Summary:**

The paper studies bounded memory second order approach. The main contribution is an algorithm, called greedy bounded memory BFGS, that obtains superlinear convergence in the non-asymptotic regime and has an explicit dependence on the memory. Experiments have been performed to verify the effectiveness of the approach.

Second order approaches like quasi newton approach obtain superlinear convergence rate (instead of linear convergence rate as gradient descent) for minimizing a $\mu$-strongly convex and $L$-second order smooth function. However, these approach typically requires quadratic memory ($d^2$, where $d$ is the dimension of the function) and relative large computation cost. The bounded memory BFGS is designed to save the memory, by storing a subset of past curvature (e.g. the last $\tau$ points and gradients). However, a major challenge in the literature is to derive superlinear convergence rate for bounded memory approach. This question is resolved in a recent paper by [Rodomanov and Nesterov 2021]. However, this result is in the asymptotic regime and no explicit dependence on the memory is shown.

The major contribution of this paper is to provide the convergence analysis in the non-asymptotic regime and the convergence rate derived has an explicity dependence on the memory. The convergence rate obtained is roughly: $(1 - \frac{\mu}{C_{\beta}dL})^{t(t+1)/2}(1-\frac{\mu}{2L})^{t_0}$ where $C_{\beta}$ is a parameter depends on the memory.

The main idea of the paper is (1) a greedy selection procedure that selects the direction that maximizes the deduction of potential function; (2) a displacement aggregation that determines the curvature to store. As far as I understand, both ideas have appeared in the literature, but combining them is quite non-trivial.

I like the presentation of this paper, and I believe the non-asymptotic result is of broad interests to optimization and OR community. I incline to acceptance, though I have to say I am not an expert and perhaps miss its overlap between literature.

**Strengths:**

The result looks very nice.

**Weaknesses:**

The paper claims an explicity dependence on memory. This dependence is hidden in a parameter $C_{\beta}$, looking at its definition, I can understand its relationship with the memory. However, this relationship is still not that explicit because it is not an explicit function of the memory.

**Questions:**

.

---

> ### Author Response · Authors · 2023-11-20
> **Response by Authors**
>
> We would like to thank the Reviewer for reviewing our manuscript and for the constructive feedback. We are glad to see the Reviewer appreciated the presentation and the results of the paper. In what follows, we provide our response and actions to your comments.
>
> **Response to Weakness**: The Reviewer is correct that the minimal relative condition number $\beta_\tau$ (or the bound $C_\beta$) embeds the role of the memory size $\tau$ in the convergence rate. Specifically, a larger $\tau$ decreases the minimal relative condition number $\beta_{\tau}$, reduces the contraction factor $1-\mu/(\beta_{\tau}dL)$, and improves the convergence rate, but increases the storage memory and computational cost; hence, yielding an inevitable trade-off between these factors.
>
> On the other hand, we agree with the Reviewer that interesting topics for future research include deriving explicit dependency of $\beta_\tau$ (or $C_\beta$) on the memory size $\tau$. We consider this work as the first stride to explore the possibility of achieving superlinear rates with limited-memory BFGS methods and analyze the inherent role of the memory size on the convergence rate. We will clarify this point in the final upload. Thank you for the comment.

---

### Author Response · Authors · 2023-11-20
**General Response to Reviewers and Area Chair**

We sincerely thank all the Reviewers and Area Chair for their time in assessing our work and providing constructive feedback, which contributed to an improved paper. Before providing a detailed response to each of the Reviewers' comments, we would like to highlight the contributions of our work:

1. We propose a novel synthesis of greedy basis selection and displacement aggregation, which maximizes the update progress per iteration while maintaining the limited memory setting. Specifically, we conduct greedy selection along a limited-memory basis [cf. (6)], which regulates variable variations within a fixed subspace. The latter restricts the count of linearly independent curvature pairs and enables the ongoing displacement aggregation to prune unnecessary curvature information, ensuring compliance with the memory limitation. The resulting algorithm, LG-BFGS, is the first to introduce such an approach.

2. We furnish a clear and explicit expression for the non-asymptotic convergence of the proposed LG-BFGS, which not only delineates the direct influence of memory size through the concept of relative condition number $\beta_\tau$ [Def. 1], but also identifies specific roles played by other factors (e.g., problem properties, initial conditions, etc.). With the boundness assumption on $\beta_\tau$, it presents an explicit superlinear convergence rate. This is the first work that characterizes the explicit effect of memory size on the convergence rate and provably shows the possibility of non-asymptotic superlinear rates in limited-memory BFGS methods.

3. We identify the exact assumption required for limited-memory BFGS methods to achieve explicit superlinear rates. Up to this paper, it was unclear if limited-memory methods could achieve non-asymptotic superlinear rates and under which conditions / assumptions this could be possible. This paper not only provides a definitive answer, but also identifies the required condition explicitly. Our assumption and results are consistent with the full-memory BFGS method, i.e., the bound reduces to $1$ and the superlinear rate becomes that of the full-memory method when the memory size increases to the feature dimension. This is the first work that establishes such a clear path from the limited-memory method to its full-memory counterpart, and provides an explicit theoretical explanation behind their performance changes along the reduction of memory size.

In summary, this paper answers three unknown questions in the literature: (i) Is it possible for limited-memory methods to achieve explicit superlinear rates? (ii) If so, which condition does it require? (iii) What is the explicit effect of memory size on the convergence rate? Therefore, we consider this work is of significance as the first stride to fill in these gaps.

---

> ### Author Response · Authors · 2023-11-23
> **General Response to Reviewers and Area Chair**
>
> As the discussion period is drawing to close, we sincerely appreciate the Reviewers for providing high-quality comments and nice suggestions, which helped strength our paper. If our responses have addressed your concerns, we humbly hope the Reviewers could revisit the score in light of the responses and revisions.
>
> Thank you again for your time and effort in reviewing our work.

---

### Meta-Review · Area_Chair_nu3b · 2023-12-09

**Metareview:**

The paper aims to establish super linear convergence guarantees for a variant of limited memory BFGS, which incorporates a combination of greedy selection and displacement aggregation.

The paper's focus is mainly on obtaining theoretical guarantees (as a result, the numerical section is less developed). However, the theory provided in the paper seems rather limited. There are concerns regarding some of the assumptions made to obtain super-linear rate. For example, the minimal relative condition number of the error matrix is assumed to be bounded. However, the error matrix is an algorithmic construction that depends on the algorithm and has no apparent connection to the underlying objective function. In other words, this amounts to making a priori assumption on the future performance of the algorithm, and not an assumption that can be verified, at least hypothetically, prior to running the algorithm. As a result, in the absence of an actual non-trivial function class that satisfies the assumption of the bounded relative condition number of the error matrix, it is rather impossible to verify the paper's claim of establishing super-linear convergence.

**Justification For Why Not Higher Score:**

The theory provided in the paper seems rather limited. There are concerns regarding some of the assumptions made to obtain super-linear rate.

**Justification For Why Not Lower Score:**

N/A

---

### Decision · Program_Chairs · 2024-01-16

Reject